# Occurrence and Determination of Carotenoids and Polyphenols in Different Paprika Powders from Organic and Conventional Production

**DOI:** 10.3390/molecules26102980

**Published:** 2021-05-17

**Authors:** Alicja Ponder, Klaudia Kulik, Ewelina Hallmann

**Affiliations:** Department of Functional and Organic Food, Institute of Human Nutrition Sciences, Warsaw University of Life Sciences, Nowoursynowska 159c, 02-776, Warsaw, Poland; klaudia_kulik@sggw.edu.pl (K.K.); ewelina_hallmann@sggw.edu.pl (E.H.)

**Keywords:** bell pepper, carotenoids, polyphenols, paprika powder, organic, conventional, basket study

## Abstract

Paprika powder is a good source of different carotenoids and polyphenols, which play a key role in preventing certain diseases (some kinds of cancer and cardiovascular diseases). They can also be used as natural food colorants. Organic production is characterized by strict rules, but products obtained in this way contain more bioactive compounds, such as carotenoids and polyphenols. The aim of this study was to measure and identify carotenoids and polyphenols in different paprika samples (sweet, hot, smoked, and chili) obtained by organic and conventional production. Quantitative and qualitative carotenoid and polyphenols analysis showed that the experimental samples contained different concentrations of these compounds.

## 1. Introduction

In recent years, consumers have been increasingly interested not only in food safety but also in its quality, including attributes such as the presence of specific compounds or the production system (e.g., organic production). Several spices such as paprika powdered spice are widely employed as a food seasoning in the world cuisines because of their organoleptic and pro-healthy properties. A valued red powdered spice is obtained from the drying and grinding of red pepper fruits of the genus *Capsicum* (Solanaceae family) [1]. Paprika powder prepared from *Capsicum annuum* is a natural source of carotenoids and polyphenols. More than ten different carotenoids occur in the fruits [2,3,4,5]. The main carotenoids identified in paprika powder are *beta*-carotene, capsorubin, capsanthin, lutein, zeaxanthin, and violaxanthin (Appendix A). Carotenoid compounds in bell pepper fruits occur more frequently in complex forms, such as ester (mono- and di-esters), than in free forms [6,7]. The group of ground bell pepper spices includes products such as sweet paprika, hot paprika, and chili pepper [8]. They can be used in food as spices, colorants, and antioxidant agents [9,10,11]. Moreover, carotenoids have healthful properties and can prevent chronic diseases such as different kinds of cancer, cardiovascular disease, type 2 diabetes, and atherosclerosis [12,13,14]. Many studies have shown that organic vegetables contain more bioactive compounds, including carotenoids, than conventionally grown vegetables [15,16,17,18]. Whereas polyphenols are often responsible for the antioxidant capacity of plant products. Polyphenolic compounds are secondary metabolites ubiquitously spread through the plant kingdom comprising more than 8000 substances with highly diverse structures (Appendix A). Over the last few years, this family of compounds has caught a lot of attention due to the recognition of their antioxidant properties and their probable role in the prevention of several diseases [19,20,21]. In paprika the most important phenolic compounds are flavonoids and phenolic acids [10]. Polyphenolic content in natural food products seems to be related to plant features such as the plant cultivar, the geographical climate conditions of their production area, and the cultivation and manufacturing practices. Consequently, phenolic profiling can be exploited as a source of analytical data to establish product classifications, as well as for the evaluation of food quality [19]. Organic production is a strict system with special rules. Artificial fertilizers as well as pesticides are forbidden. Only natural methods of production are allowed (animal and green manure, compost). For plant protection, pheromone traps, natural predators, and sticky boards are used. Such production conditions stimulate plants to produce more bioactive compounds. There is scarce information about the identification of individual carotenoids in organic products prepared from *Capsicum annuum* fruits. Only two previous experiments have been conducted to determine carotenoids in fresh and pickled organic bell pepper fruits [22,23]. In the literature, there is a limited amount of information about the qualitative and quantitative composition of polyphenolic compounds in the bell pepper products. Only a few experiments with polyphenols content have been done with bell pepper products [10,24,25]. While the aim of this study was to conduct quantitative and qualitative identification of carotenoids and polyphenols in four different organic and conventional bell pepper products: sweet, hot, smoked, and chili pepper powders.

## 2. Results

The obtained results showed that conventional paprika powder was characterized by a higher total carotenoid content than organic paprika powder (*p* > 0.0001). Among the examined paprika products in both production systems (organic and conventional), smoked and chili pepper contained significantly more total carotenoids than sweet and hot paprika (*p* < 0.0001). *Beta*-carotene and capsorubin concentrations were significantly higher in organic samples (*p* < 0.0001 and *p* < 0.0001) than in conventionally produced samples (Table 1). Chili pepper powder contained significantly more *beta*-carotene and capsorubin (*p* = 0.0013 and *p* < 0.0001) than the rest of the examined paprika samples. According to the data presented in Table 1, conventional paprika samples were characterized by a significantly higher concentration of *cis*-*beta*-carotene (*p* < 0.0001). Smoked paprika samples contained significantly more of this carotene than the rest of the examined samples (*p* < 0.0001). Was noticed that organic paprika powder samples contained significantly more *alpha*-carotene than conventional samples (*p* = 0.0363). A significantly higher concentration of *alpha*-carotene (*p* < 0.0001) was observed in smoked paprika samples than in the other samples (Table 1). In the xanthophyll group, conventional paprika powders contained significantly more cryptoxanthin (*p* < 0.0001). Among individual samples, chili pepper was characterized by a significantly higher cryptoxanthin concentration (*p* < 0.0001). Was observed that the concentrations of cryptoflavin, *beta*-cryptoflavin and lutein depended mostly on the production system. Organic samples were characterized by a significantly higher content of those xanthophylls than conventional samples (*p* < 0.0001, *p* < 0.0001 and *p* < 0.0001, respectively) (Table 2). No effect of products as a second factor was observed. In the case of beta-cryptoxanthin, conventional paprika powdered samples contained significantly more of this colorant (Table 2). The effect of the production system was observed only in the case of zeaxanthin content. Organic samples contained significantly more zeaxanthin than conventional samples (*p* < 0.0001). In both production systems, smoked paprika samples contained significantly more zeaxanthin than the rest of the experimental samples (*p* < 0.0001).

Moreover, the obtained results also showed that conventional paprika powder was characterized by a higher content of capsaicin than organic paprika powder (*p* = 0.0002) (Table 3). Whereas, among the examined paprika products in both production systems (organic and conventional), chili pepper contained significantly the most capsaicin (*p* < 0.0001). In addition, the effect of the production system on the content of polyphenols was observed (Table 3 and Table 4). Was noticed that organic paprika powder samples contained significantly more caffeic acid (*p* = 0.0002), ferulic acid (*p* < 0.0001), total flavonoids (*p* = 0.028), and quercetin-3-*o*-rutinoside (*p* < 0.0001) than conventional samples (Table 4). While, conventional paprika powders contained significantly more total polyphenols (*p* < 0.0001), total phenolic acids (*p* < 0.0001), gallic acid (*p* < 0.0001), chlorogenic acid (*p* < 0.0001), myricetin (*p* < 0.0001), quercetin (*p* = 0.027), quercetin-3-*o*-glucoside (*p* = 0.0078), and kaempferol (*p* < 0.0001). Was also observed that the concentrations of polyphenols depended on the type of product. Conventional smoked paprika and chili pepper powder contained significantly the most total polyphenols (*p* < 0.0001), total phenolic acids (*p* < 0.0001), and chlorogenic acid (*p* < 0.0001) than the rest of the examined paprika samples. Among the examined paprika products in both production systems (organic and conventional), smoked paprika contained significantly the most quercetin-3-*o*-glucoside (*p* < 0.0001). 

PCA showed a high and significant overall variation between experimental objects, and 85.79% of the variation was explained by PC1 and PC2 (Figure 1). A degree of dependence between the conventional and organic samples was described for most identified carotenoids. For SmBPC and SmBPO, the carotenoids were zeaxanthin (Zea), *cis-beta*-carotene (c-b-C), *alpha*-carotene (a-C), *cis*-zeaxanthin (c-Zea), and *alpha*-cryptoxanthin (a-Cry-X). For ChPC and ChPO, they were total carotenoids (TC), *beta*-cryptoxanthin (b-Cry-X), *beta*-carotene (b-C), cryptoxanthin (Cry), capsorubin (Cap-R), and lutein (Lut). Only in the cases of HBPC and HBPO and of SBPO and SBPC was the effect of cryptoflavin observed (Figure 1). It is worth noting that the examined experimental objects were located in completely different parts of the chart, which showed a higher variation between samples.

## 3. Discussion

The quality of natural food products is an issue of great interest in our society, especially when dealing with organic products. Several health-promoting properties of those have been partly attributed to the presence of bioactive compounds, which also play an important role in food sensorial and functional properties. Paprika, or chili pepper, which is a red seasoning powder with a characteristic flavor obtained from the drying and grinding of certain varieties of red peppers (*Capsicum annuum* L.), is one of the most widely used food colorants in both culinary and industrial applications. The intense and characteristic red color of paprika is due to its high carotenoid pigment content produced during fruit ripening, that apart from conferring color to food provides important health benefits [19]. Bell pepper fruits accumulate carotenoids, including carotenes and xanthophylls, at the time of maturity. Products (dry powders) prepared from different kinds of bell peppers contain different levels of carotenoids. In the experimental trials with paprika powders, eleven individual carotenoids were identified (Table 1 and Table 2, Figure 1). The dominant carotene occurring in all experimental pepper samples was *alpha*-carotene. The concentration range was from 22.34 mg/100 g DW to 35.59 mg/100 g DW. A similar range of α-carotene was reported for *Capsicum annuum* and *Capsicum chinense* species in the range from 2.22 mg/100 g DW to 21.27 mg/100 g DW [26]. The processing of paprika fruits can change the level of carotenoids in the final product. Paprika powders after smoking contained almost twice the total carotenoids (Table 1). Similar results were observed in experiments using different types of wood for the smoking of paprika fruits. After processing, the total carotenoid content increased compared to that of fresh fruits [27]. Organic plant production influences carotenoid content in bell pepper fruits. According to the literature, organic bell pepper contains 7.5 mg/100 g DW of *beta*-carotene and conventional only 6.3 mg/100 g DW [22]. The concentration of *beta*-carotene in bell pepper fruits is affected by species. Sweet bell pepper cultivars contain less *beta*-carotene than hot cultivars [28]. The experimental trials was noticed similar differences (Table 1). The balance between *beta*-carotene and its isomer *cis-beta*-carotene is not stable and can change over time during fruit processing. After the processing (pickling) of organic bell pepper fruits, the *cis*-isomer form increased [23]. A similar relationship was observed in the present experiment, especially in organic products (Table 1). The xanthophyll content in paprika powder products is dependent mostly on the drying method used for product preparation. The concentrations of cryptoxanthin and its isomer *beta*-cryptoxanthin were affected by the smoking process. After that processing, the levels of both xanthophylls were increased in the final product (Table 2). Similar observations were made by Topuz et al. (2011) [29]. The concentration of cryptoxanthin in paprika products ranged from 0.06–0.22 mg/100 g DW [30]. Similar results were observed in experimental trials (Table 2). The smoking process influences the concentrations of zeaxanthin and *cis*-zeaxanthin in paprika powder products. Was observed that after smoking, the concentrations of both isomers increased 1.5-fold compared to those in sweet paprika samples. A similar effect was reported in experiments with different drying and preservation methods of paprika [31]. Paprika powder is a good source of lutein. The experimental trials was found significantly higher concentrations of lutein in organically produced samples than in conventional samples (Table 2). It seems that after the smoking process, the level of lutein increased in the examined paprika powder samples. Generally the smoked paprika samples contained usually more carotenoids than the other experimental samples. Therefore, it can be assumed that carotenoids become more available during the smoking process. Thus, the smoking process increases the availability of carotenoids, as is the case with the use of various processing processes using high temperature [32,33].

Carotenoids are found in the most commonly consumed fruits and vegetables with a strong orange, red, yellow, or green color (Appendix A) [24,32,33,34,35,36,37,38,39,40,41,42,43,44,45]. Apart from paprika, carrots are one of the richest dietary sources of beta-carotene and still high but lower extent alpha-carotene [34,35]. The same major carotenoid is representative for spinach [35]. In our study the dominant carotene in all pepper samples was alpha-carotene. The main carotenoid in fresh tomato fruits is lycopene, but Mendelova et al. (2013) [36] show that processes like heating and drying tomatoes lead to an increase in the content of carotenoids at the level from 85.71–169.85 mg/100 g dry mass vs. 55.73–106.89 mg/100g dry mass before processes. Lutein was determined as major carotenoid in spinach, beside beta-carotene [35,37,38]. Whereas all our bell pepper products, in both production systems (organic and conventional) were good source of lutein. The total carotenoid content of fresh produce such as peppers varies greatly due to many factors such as variety, type, growing conditions, etc. Wall et al. (2001) shown that total carotenoids content in red jalapeno pepper and different species, cultivars, and types of fresh *Capsicum* fruit (*C. annum*-Cayenne (red), *C. chinense*-Habanero (red) and *C. frutescens*-Tabasco) are relatively high but quite varied, (range 23.4–43,8 mg/100 g DW, 37.9–50.5 mg/100 g DW, 34.8–44,6 mg/100 g DW, and 56.2–84.9 mg/100g DW, respectively) [37]. The results we obtained, also confirm, that in all bell pepper powders content of total carotenoids in both production systems are different (statistically significant). It should be noted that converting the total carotenoid content to the dry mass of the product may generate much higher values compared to FW. According to the literature, *C. annum* Cayenne (red) and ***C.***
*frutescens* Tabasco contains from 357–399 and 133.4–199.9 mg/100 g DW, respectively [37,39,46]. Literature data reports that the beta-carotene content of paprika is higher in red and brown fruit compared to orange or yellow. Park et al. (2014), in his study confirm this rule, red paprika contains the highest level of beta-carotene, 19.34 mg/100 g DW, while orange 9.74 mg/100 g DW and yellow fruit 3.19 mg/100g DW [38,47]. These higher values compared to our results are related to the sum of b -carotene isomers.

In our study, among the examined paprika products in both production systems (organic and conventional), chili pepper contained significantly the most capsaicin (1447.17 mg/100 g DW in organic chili pepper, 1452.87 mg/100 g DW in conventional chili pepper) (Table 3). While, in another study, capsaicin content of seven Indian peppers varieties/accessions from *Capsicum annuum*(CA 97, CCH, K1, KTPL19, *Arka Abhir,* and *Bayadagi Kaddi*) and *C. frutescens* (CF1) species were determined [48]. Based on their pungency value, all the chilli accession/varieties (CA 97, CCH, K1, and CF 1) were classified as highly pungent peppers. The accession CF1 showed the highest concentration of capsaicin (445.00 mg/100 g DW) and *Arka Abhir* variety showed the lowest capsaicin concentration (29.00 mg/100 g DW). The differences in capsaicin content presented in the different peppers cultivars can be exploited for breeding cultivars with improved nutritional qualities and as a potential source for capsaicin. Other researchers determined capsaicin content of several of capsicum fruit [24]. The samples were collected from different area in Indonesia which consisted of twelve types of edible *Capsicum*. The result of analysis showed that green paprika, yellow paprika, and red paprika did not contain capsaicin, while Highest capsaicin concentration was obtained in green rawit chili (cayenne) and followed by red rawit chili. In a similar experiment, the comparative analysis of capsaicin content in pepper (*Capsicum annuum*) grown in conventional and organic agricultural systems was carried out [25]. Pepper genotypes *Strumicka Kapija, Strumicka Vezena, Piran, Zupska Rana, Duga Bela,* and *Kurtovska Kapija* were material of this study. Organically grown pepper fruits were characterized with higher capsaicin content than the conventional one. The genotype *Strumicka Vezena* was characterized with the highest capsaicin content in both cultivation systems (957.00 mg/100 g DW in organic, 722.00 mg/100 g DW in conventional). However, in the experimental trials, conventional pepper contained more capsaicin than organic ones.

Along with carotenoids, polyphenolic compounds are also important bioactive compounds present in paprika. These compounds act as antioxidants, and are important in plant defense responses, having an impact in the resulting fruit quality [19]. Polyphenols are also responsible for the antioxidant capacity of paprika. Was noticed that conventional paprika powders contained significantly more total polyphenols (432.78 mg/100 g DW) than organic ones (264.82 mg/100 g DW). Was also observed that the concentrations of polyphenols depended on the type of product. Conventional smoked paprika and chili pepper powder contained significantly the most total polyphenols than the rest of the examined paprika samples (611.17 mg/100 g DW and 798.45 mg/100 g DW). In another study the content of polyphenols was assessed in two common culinary spices—paprika spices (12, ground powder spices) and pepper spices (20, unground and ground, black, green, white, and colored spices) of Czech, Austrian, and Slovak producers [10]. For paprika the total polyphenols content ranged from 1467 mg GAE/100 g to 2878 mg GAE/100 g. There was only weak connection between the pungency of the spices and the polyphenolic amount, the hotter samples of paprika spices have slightly higher values of total polyphenols than sweet types. Total polyphenols content for pepper spices was assessed in the range of 1203 mg GAE/100 g to 2288 mg GAE/100 g. Generally, paprika spices contained more polyphenols than pepper spices. The obtained results about polyphenols content can be compared with the results of other researchers, where the total phenolic content in Mexican peppers was in the range 20–782 mg/100 g [49]. In the next study the total phenols and flavonoids content of extracts of paprika powder were investigated [50]. Paprika presented a content of the total phenols 675–1360 mgAG//100g DW and flavonoids 121.36–130.20 mg quercitin/100 g DW.

Cieślik at al. (2006) reported very high total polyphenol level for tomato, carrot, italian cabbage, onion, and broccoli, for about three to sixfold, compared to our results (Appendix A) [51]. Onion is the riches vegetable. Among vegetables and fruits, onions, especially red onion and shallots are the richest in quercetin. According to Chu et al. sweet potato is good source of flavonols: myricetin and quercetin, while in spinach, all of the above-mentioned flavonols, also kaempferol content should be noticed (Appendix A) [52].

## 4. Materials and Methods

### 4.1. Analytical Material Preparation

Selected paprika samples were purchased from organic and conventional shops in Warsaw (Poland). Experiment represent “basket study” evaluation. In those kind of treatment all objects are purchase in shops. For experimental purposes, four different kinds of products were chosen: sweet paprika (SBP), hot paprika (HBP), smoked paprika (SmBP), and chili pepper (ChP), each obtained by organic production (O) and conventional production (C.). Each experimental combination (kind of product) was represented by 6 boxes (100 g net weight) from different delivery lots. One box was treated as one replicate (n = 6).

### 4.2. Chemicals

Acetone (HPLC grade) from Sigma-Aldrich (Poland, Warsaw); diethyl ether (HPLC grade) from Merck, Poland, Warsaw; hexane (HPLC grade) from Sigma-Aldrich (Poland, Warsaw); methanol (HPLC grade) from Sigma-Aldrich; carotenoid standards (HPLC grade 99.5–99.9% pure) from Sigma-Aldrich: *beta*-carotene, lutein, zeaxanthin, and magnesium carbonate (pure) from ChemPur (Warsaw, Poland); sodium carbonate (analytical grade) from ChemPur; sodium chloride (analytical grade) from Sigma Aldrich and sodium peroxide (analytical grade) from ChemPur.

### 4.3. Carotenoid Analysis

Carotenoid saponification and quantitative and qualitative analysis were performed by HPLC [53]. A product sample (100 mg) was weighed into a laboratory glass flask with acetone (purity for HPLC) and 10 milligrams of MgCO_3_. Samples were extracted on a rotary platform with cover (protection against the light) for 12 h, and the glass flasks were covered by silver aluminum foil (against light protection). After 12 h, carotenoids were rinsed and make pure with a Schott funnel (P-4, 19/26) using. Next, to each sample NaCl (30%) was added and washed out by acetone from the sample. Diethyl ether with Na_2_SO_4_ were added to the sample. After that, mixture of methanol and sodium hydroxide (40%) was added, to start saponification process which was continue through 12 h in the dark condition (with a rotary shaker). Next samples were washed out with deionized water (H_2_O) and transferred into a vacuum flask. In the next steep, (C_2_H_5_)_2_O was evaporated at 38 °C. The sediment, consisting of dry carotenoid layers, were dissolved in n-hexane (5 mL) and analysed by HPLC. For the determination of carotenoids, a Shimadzu HPLC-set (Shim-pol, Warsaw, Poland) with two LC-20AD pumps, a CMB-20A set controller, a SIL-20AC autosampler, an SPD-20AV visible light detector with spectrum identification, a CTD-20A oven and a Max-RP 80A column (size: 4.6 × 250 mm) was used. Mobile phase was prepared from mixture of acetone and n-hexane (5:95) and used as gradient phase. Time flow rate: 1.5 mL min^−1^; wavelength range was 445–480 nm. For purpose of carotenoids compounds qualitative identification, an external standard in the form of *beta*-carotene, lutein, and zeaxanthin (Sigma-Aldrich, Warsaw, Poland) of 99.9% purity was used. For the quantitative identification of individual carotenoids, coefficients were calculated (on the base of beta-carotene external standard) with the formula (Equation (1)):F_i/β-carotene_ = f_i_/f_β-carotene_(1)
f_i_ represents the recognized carotenoid, f_β−carotene_ describe pure beta-carotene, and F_i/β−carotene_ is the ratio of the identified carotenoid to pure *beta*-carotene. More details are gives in reference [46]. Chromatograms of the identified carotenoid compounds are presented in Figure 2. Identified carotenoids in paprika powder are: *beta*-carotene, *cis-beta*-carotene, *alpha*-carotene, capsorubin, cryptoxanthin, cryptoflavin, *beta*-cryptoflavin, *beta*-cryptoxanthin, lutein, zeaxanthin, and *cis*-zeaxanthin.

### 4.4. Polyphenols Analysis

Polyphenols also were performed by HPLC method using the HPLC [23]. Firstly, 100 mg of freeze-dried fruit sample was put into a plastic tube, then 80% methanol was added, mixed thoroughly by vortex, and incubated in an ultrasonic bath (temperature 30 °C, time 15 min). Then the samples were centrifuged at the speed of 3520× *g*, 900 µL of supernatant was collected from the plastic tube and re-centrifuged at the speed of 42,673.1× *g*. Then 500 µL of supernatant was injected to the HPLC vials. Synergi Fusion-RP 80i column (250 × 4.60 mm) was used for the analysis. Polyphenols were separated under gradient conditions with a flow rate of 1 mL min^−1^ by applying an aqueous solution of 10% (*v*/*v*) acetonitrile (phase A) and 55% (*v*/*v*) acetonitrile (phase B), both acidified by ortho-phosphoric acid to pH 3.0. Time of the analysis: 38 min. The phases changed as follows: 1.00–22.99 min 95% phase A and 5% phase B, 23.00–27.99 min 50% phase A and 50% phase B, 28.00–28.99 min 80% phase A and 20% phase B, 29.00–38.00 min 95% phase A and 5% phase B. Wavelengths: 250 nm for flavonoids, and 370 nm for hydroxybenzoic and hydroxycinnamic acids. The compounds were identified based on Fluka and Sigma Aldrich external standards with purity of 99.5%. The total polyphenol content was calculated as the sum of all identified individual polyphenolic compounds, i.e., gallic acid, chlorogenic acid, caffeic acid, p-coumaric acid, ferulic acid, quercetin-3-***o***-rutinoside, myricetin, quercetin, quercetin-3-***o***-glucoside, and kaempferol (Appendix A).

### 4.5. Statistical Analysis

All results were statistically elaborated. Analysis of variance with two factors was performed with Tukey’s test (*p* = 0.05) using Statgraphics^®^ Centurion 15.2.11.0 (StatPoint Technologies, Inc., Warranton, VA, USA). In the described examination, two factors were analysed: the production method (organic and conventional) and the kind of product (sweet, hot, smoked, and chili peppers). The number of samples per system of production was n = 12, and the number of samples per product was n = 6. A lack of statistically significant difference (*p* > 0.05) is described in the tables as N.S. Different letters within a row indicate statistically significant differences at the level *p* < 0.05. PCA (Principal Component Analysis) was performed with XLSTAT Software package (XLSTAT, 2020, New York, NY, USA).

## 5. Conclusions

In conclusion, many factors, such as the type of product, processing method and type of production, appeared to have important effects on the quality and quantity of carotenoids and polyphenols in paprika powders. It is worth noting that many results highlighted the impact of different types of fresh bell peppers on carotenoid and polyphenols status, but not the difference between organic and conventional paprika products. The choice of an appropriate paprika preparation technique has a significant impact on the carotenoid and polyphenols contents of the products.

## Figures and Tables

**Figure 1 molecules-26-02980-f001:**
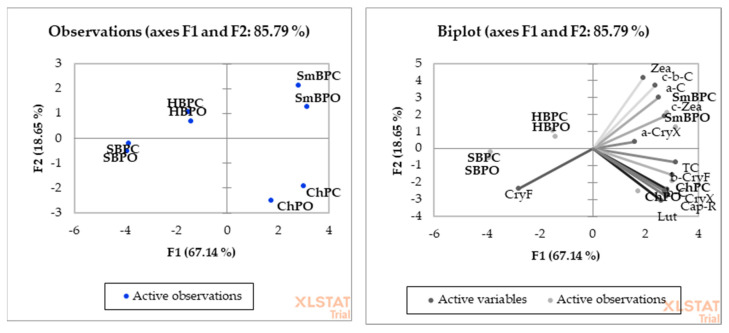
PCA analysis showing the relationship between the chemical composition carotenoids content in organic (O) and conventional (C.) paprika products. (TC) total carotenoids, (b-C) beta-carotene, (c-b-C) cis-beta-carotene, (a-C) alpha-carotene, (CryX) cryptoxanthin, (CryF) cryptoflavin, (b-CryF) beta-cryptoflavin, (a-CryX) beta-cryptoxanthin, (Lut) lutein, (Zea) zeaxanthin, (c-Zea) cis-zeaxanthin, (Cap-R) capsorubin.

**Figure 2 molecules-26-02980-f002:**
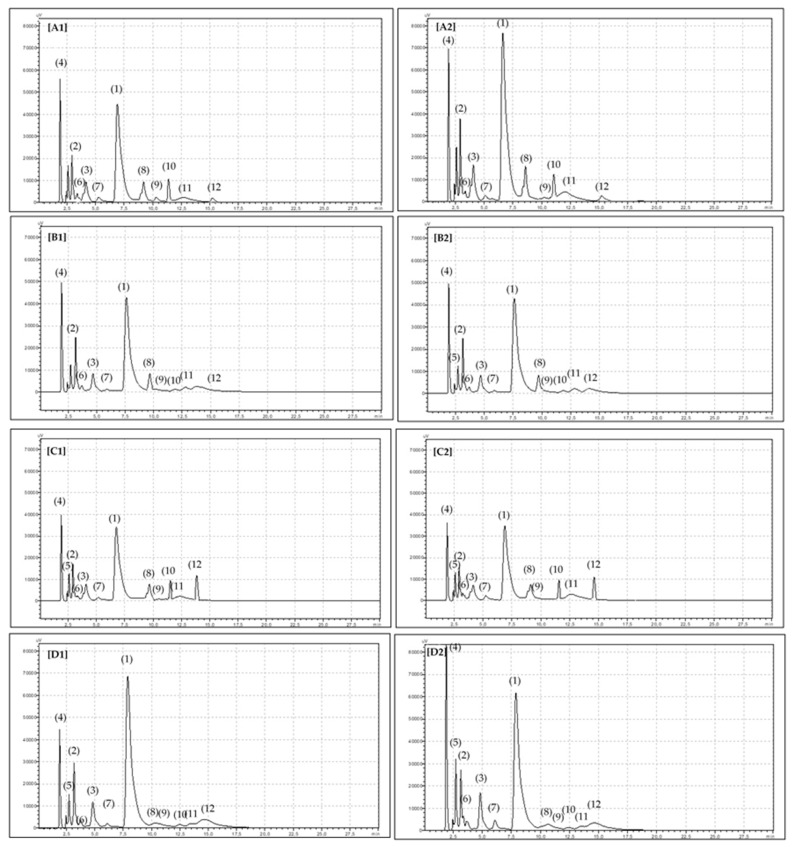
The picture of identified carotenoids in paprika powder: sweet organic [**A1**]; sweet conventional [**A2**]; hot organic [**B1**]; hot conventional [**B2**]; chili orgnic [**C1**]; chili conventional [**C2**]; smoked organic [**D1**]; smoked conventional [**D2**]. (1) *beta*-carotene, (2) *cis-beta*-carotene, (3) *alpha*-carotene, (4) capsorubin, (5) cryptoxanthin, (6) cryptoflavin, (7) *beta*-cryptoflavin, (8) *beta*-cryptoxanthin, (10) lutein, (11) zeaxanthin, (12) *cis*-zeaxanthin.

**Table 1 molecules-26-02980-t001:** The content of total carotenoids and individual identified carotenes in different bell pepper products (in mg 100 g DW).

Production System	Product	Total Carotenoids	*beta*-Carotene	*cis-beta*-Carotene	*alpha*-Carotene	Capsorubin
Organic	Sweet Paprika	52.4 5± 0.27 c	4.24 ± 0.09 c	1.54 ± 0.02 d	22.34 ± 0.25 c	2.30 ± 0.02 d
	Hot Paprika	66.63 ± 1.11 b	5.23 ± 0.10 b	4.15 ± 0.0 c	29.87 ± 1.01 a	3.12 ± 0.23 c
	Smoked Paprika	91.25 ± 0.57 a	6.55 ± 0.15b	6.45 ± 0.16 a	35.59 ± 0.40 a	4.93 ± 0.16 b
	Chili Pepper	87.34 ± 1.29	7.35 ± 0.15 a	3.31 ± 0.08	25.30 ± 0.28 b	5.64 ± 0.13 a
	Mean Value	74.42 ± 3.25 B	5.85 ± 0.25 A	3.86 ± 0.36 B	28.27 ± 1.06 A	3.99 ± 0.28 A
Conventional	Sweet Paprika	55.67 ± 0.22 c	3.43 ± 0.07 c	2.43 ± 0.01 d	22.96 ± 0.26 c	2.54 ± 0.02 d
	Hot Paprika	66.18 ± 0.73 b	4.70 ± 0.11 c	5.57 ± 0.08 b	25.85 ± 0.21 b	2.50 ±0.02 d
	Smoked Paprika	91.20 ± 0.55 a	5.72 ± 0.10 b	8.28 ± 0.1 a	32.08 ± 0.33 a	3.92 ± 0.14 b
	Chili Pepper	101.67 ± 0.69 a	7.43 ± 0.06 a	5.09 ± 0.05 b	29.15 ± 0.29 a	5.76 ± 0.18 a
	Mean Value	78.68 ± 3.79 A	5.32 ± 0.30 B	5.35 ± 0.43 A	27.51 ± 0.71 B	3.68 ± 0.28 B
*p*-Value	Production System (P)	<0.0001	<0.0001	<0.0001	0.0363	0.0048
	Product (Pr)	<0.0001	<0.0001	<0.0001	<0.0001	<0.0001
	Interaction (PxPr)	<0.0001	0.0013	<0.0001	<0.0001	<0.0001

Data are presented in tables are the mean ± SE with ANOVA *p*-value; Average values in columns with the different letter are statistically significant at the 5% level of probability; N.S. not significant statistically; (n) = 6.

**Table 2 molecules-26-02980-t002:** The content of individual identified xanthophylles in different bell pepper products (in mg 100 g DW).

Production System	Product	Cryptoxanthin	Cryptoflavin	*beta*-Cryptoflavin	*beta-Cryptoxanthin*	Lutein	Zeaxanthin	*cis*-Zeaxanthin
Organic	Sweet Paprika	14.88 ± 0.08 c	1.08 ± 0.03 a	0.30 ± 0.01 a	0.23 ± 0.00 a	3.12 ± 0.11 a	0.45 ± 0.01 c	1.98 ± 0.01
	Hot Paprika	17.07 ± 0.23 b	0.92 ± 0.03 a	0.44 ± 0.01 a	0.28 ± 0.00 a	2.93 ± 0.06 a	0.54 ±0.00 b	2.10 ± 0.01 bc
	Smoked Paprika	27.16 ± 0.20	0.76 ± 0.01 a	0.65 ± 0.01 a	0.53 ± 0.01 a	5.22 ± 0.10 ba	0.77 ±0.01 a	2.63 ± 0.02 a
	Chili Pepper	34.80 ± 1.37 ab	0.86 ± 0.01 a	0.71 ± 0.01 a	0.59 ± 0.02 a	6.14 ± 0.14 a	0.46 ±0.01 c	2.18 ± 0.01 b
	Mean Value	23.48 ± 1.67 B	0.91 ± 0.03 A	0.52 ± 0.03 A	0.41 ± 0.03 B	4.35 ± 0.28 A	0.55 ±0.03 A	2.22 ± 0.05 A
Conventional	Sweet Paprika	17.95 ± 0.10 c	0.99 ± 0.02 a	0.20 ± 0.00 a	0.31 ± 0.01 a	2.53 ± 0.02 a	0.32 ±0.01 d	2.01 ± 0.00 c
	Hot Paprika	20.49 ± 0.15 b	0.84 ± 0.01 a	0.34 ± 0.01 a	1.08 ± 0.13 a	2.41 ± 0.39 a	0.49 ±0.01 c	1.91 ± 0.01 d
	Smoked Paprika	31.73 ± 0.17 b	0.69 ± 0.02 a	0.58 ± 0.01 a	0.71 ± 0.02 a	4.02 ± 0.16 a	0.74 ±0.01 a	2.71 ± 0.02 a
	Chili Pepper	43.57 ± 0.62 a	0.83 ± 0.01 a	0.68 ± 0.02 a	0.86 ± 0.01 a	5.50 ± 0.16 a	0.40 ±0.02 cd	2.40 ± 0.01 b
	Mean Value	28.43 ± 2.08 A	0.84 ± 0.02 B	0.45 ± 0.04 B	0.74 ± 0.14 A	3.61 ± 0.28 B	0.49 ± 0.03 B	2.26 ± 0.07 A
*p*-Value	Production System (P)	<0.0001	<0.0001	<0.0001	0.027	<0.0001	<0.0001	N.S.
	Product (Pr)	<0.0001	<0.0001	<0.0001	N.S.	<0.0001	<0.0001	<0.0001
	Interaction (PxPr)	0.0001	N.S.	N.S.	N.S.	N.S.	0.0042	0.0412

Data are presented in tables are the mean ± SE with ANOVA *p*-value; Average values in columns with the different letter are statistically significant at the 5% level of probability; N.S. not significant statistically; (n) = 6.

**Table 3 molecules-26-02980-t003:** The content of total polyphenols, total phenolic acids, capsaicin, and individual identified phenolic acids in different bell pepper products (in mg 100 g DW).

Production System	Product	Capsaicin	Total polyphenols	Total Phenolic Acids	Gallic	Chlorogenic	Caffeic	*p*-Coumaric	Ferulic
Organic	Sweet Paprika	242.67 ± 8.58 c	226.44 ± 6.51 c	198.29 ± 6.38 c	102.58 ± 7.24 b	66.98 ± 4.45 b	11.16 ± 1.31 c	14.59 ± 0.88 c	2.98 ± 0.23 d
	Hot Paprika	731.82 ± 15.96 b	275.50 ± 5.45 c	243.49 ± 6.99 b	85.03 ± 4.59 c	58.44 ± 1.70 b	12.09 ± 0.03 c	73.99 ± 6.26 a	13.95 ± 0.37 a
	Smoked Paprika	140.50 ± 2.91 c	323.69 ± 11.62 b	196.68 ± 9.55 c	61.37 ± 2.07 d	66.96 ± 9.70 b	41.82 ± 4.97 b	15.32 ± 1.48 c	11.20 ± 0.19 a
	Chili Pepper	1447.17 ± 79.18 a	233.65 ± 6.09 c	190.27 ± 4.80 c	79.46 ± 4.47 c	33.23 ± 0.82 c	65.45 ± 1.82 a	8.61 ± 0.39 d	3.52 ± 0.16 c
	Mean Value	640.54 ± 150.49 B	264.82 ± 11.86 B	207.18 ± 7.08 B	82.11 ± 4.91 B	56.40 ± 4.82 B	32.63 ± 6.67 A	28.13 ± 7.85 A	7.91 ± 1.38 A
Conventional	Sweet Paprika	852.43 ± 32.15 b	168.88 ± 2.53 d	123.77 ± 1.20 d	62.67 ± 2.75 d	33.79 ± 1.66 c	4.70 ± 0.16 d	13.83 ± 0.79 c	8.79 ± 0.36 b
	Hot Paprika	782.96 ± 58.80 b	152.64 ± 11.27 d	122.17 ± 12.27 d	68.30 ± 10.62 d	10.55 ± 0.84 d	5.47 ± 0.14 d	35.04 ± 1.18 b	2.82 ± 0.16 d
	Smoked Paprika	119.23 ± 5.52 c	611.17 ± 35.67 a	555.67 ± 37.20 a	107.12 ± 1.96 b	360.38 ± 37.44 a	42.42 ± 3.68 b	41.06 ± 0.38 b	4.69 ± 0.15 c
	Chili Pepper	1452.87 ± 21.92 a	798.45 ± 12.57 a	725.71 ± 10.40 a	204.21 ± 1.85 a	474.03 ± 7.15 a	35.67 ± 2.49 b	8.16 ± 0.11 d	3.65 ± 0.26 c
	Mean Value	801.87 ± 137.92 A	432.78 ± 81.44 A	381.83 ± 77.38 A	110.58 ± 16.61 A	219.69 ± 59.01 A	22.06 ± 5.07 B	24.52 ± 4.01 A	4.99 ± 0.67 B
*p*-Value	Production System (P)	0.0002	<0.0001	<0.0001	<0.0001	<0.0001	0.0002	n.s.	<0.0001
	Product (Pr)	<0.0001	<0.0001	<0.0001	<0.0001	<0.0001	<0.0001	<0.0001	<0.0001
	Interaction (PxPr)	<0.0001	<0.0001	<0.0001	<0.0001	<0.0001	<0.0001	<0.0001	<0.0001

Description as in Table 1 and Table 2.

**Table 4 molecules-26-02980-t004:** The content of flavonoids and individual identified flavonoids in different bell pepper products (in mg 100 g DW).

Production System	Product	Total Flavonoids	Quercetin-3-*o*-Rutinoside	Myricetin	Quercetin	Quercetin-3-*o*-Glucoside	Kaempferol
Organic	Sweet Paprika	28.14 ± 0.30 c	5.95 ± 0.63 c	4.11 ± 0.09 c	2.73 ± 0.26 b	13.49 ± 0.60 c	1.87 ± 0.01 c
	Hot Paprika	32.00 ± 1.55 c	5.08 ± 0.22 c	3.68 ± 0.12 cd	2.52 ± 0.12 b	19.05 ± 1.38 b	1.67 ± 0.01 c
	Smoked Paprika	127.00 ± 5.97 a	67.11 ± 5.86 a	4.15 ± 0.03 c	2.35 ± 0.03 b	49.68 ± 0.36 a	3.70 ± 0.13 a
	Chili Pepper	43.38 ± 3.15 c	20.16 ± 2.13 bc	5.02 ± 0.20 b	2.87 ± 0.10 b	11.97 ± 0.97 c	3.37 ± 0.12 a
	Mean Value	57.63 ± 11.80 A	24.57 ± 7.46 A	4.24 ± 0.15 B	2.62 ± 0.09 B	23.55 ± 4.45 B	2.65 ± 0.26 B
Conventional	Sweet Paprika	45.10 ± 1.50 b	2.16 ± 0.19 d	7.21 ± 0.08 a	3.10 ± 0.05 a	27.74 ± 1.61 b	4.90 ± 0.17 a
	Hot Paprika	30.46 ± 2.85 c	6.39 ± 0.43 c	4.40 ± 0.27 c	3.75 ± 0.07 a	13.01 ± 2.31 c	2.90 ± 0.05 b
	Smoked Paprika	55.51 ± 3.04 b	5.37 ± 0.31 c	4.01 ± 0.05 c	2.61 ± 0.13 b	41.02 ± 2.83 a	2.49 ± 0.02 b
	Chili Pepper	72.74 ± 3.81 a	34.54 ± 3.92 b	4.34 ± 0.09 c	2.02 ± 0.02 b	28.94 ± 0.66 b	2.90 ± 0.06 b
	Mean Value	50.95 ± 4.68 B	12.11 ± 3.89 B	4.99 ± 0.38 A	2.87 ± 0.19 A	27.68 ± 3.04 A	3.30 ± 0.27 A
*p*-Value	Production System (P)	0.028	<0.0001	<0.0001	0.027	0.0078	<0.0001
	Product (Pr)	<0.0001	<0.0001	<0.0001	0.0005	<0.0001	<0.0001
	Interaction (PxPr)	<0.0001	<0.0001	<0.0001	<0.0001	<0.0001	<0.0001

Description as in Table 1 and Table 2.

## Data Availability

Data will be made available upon reasonable request by authors Alicja Ponder and Ewelina Hallmann.

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
