# Peer review of "Occurrence and Determination of Carotenoids and Polyphenols in Different Paprika Powders from Organic and Conventional Production"

_molecules, 2021, doi:10.3390/molecules26102980_

Round 1
Reviewer 1 Report
In this version of the manuscript, the authors have incorporated the minor suggestions made in the previous revision and I suggest that the article could be accepted in its current state.
Author Response
Reviewer no. 1
Comment 1: “In this version of the manuscript, the authors have incorporated the minor suggestions made in the previous revision and I suggest that the article could be accepted in its current state.”
Authors’ response: Thank you very much for the positive evaluation of our paper and for your recommendation to publish our manuscript in the “Molecules” journal.
Reviewer 2 Report
The authors have improved a lot of the work.
Nevertheless, I believe that it is not particularly innovative, and above all, the data obtained cannot be reproduced. The authors do not provide clear indications regarding the biological starting material.
Pedoclimatic, cultural conditions, and processing methods influence the content of phytochemicals. Therefore it is necessary to provide these indications to give credibility to the conclusions.
Author Response
Reviewer no. 2
Comment 1: “The authors have improved a lot of the work.
Nevertheless, I believe that it is not particularly innovative, and above all, the data obtained cannot be reproduced. The authors do not provide clear indications regarding the biological starting material.
Pedoclimatic, cultural conditions, and processing methods influence the content of phytochemicals. Therefore it is necessary to provide these indications to give credibility to the conclusions.”
Authors’ response: Authors agree with Reviewer suggestion. Before (in Review round no. 1) we try to show, that a many articles published in hi-score and MDPI journals based on material purchased from local markets. Presented experiment is "basket study" type. On the other hand species producers are obligatory and had always the same drying and processing methods. As a proof Authors present other experiment "basket study" type with as well drying herbs purchased from local market. One different is, that Authors use two years of experiment. Presented results are a very similar in both year (in case of polyphenols content). So is possible to obtain repeatability of results.
Of course Authors are appreciate to Reviewer for his pointing and in the next planned experiment with herbs and spices with “basket study” experiment more than one year will be used.
Hallmann E., Sabała P. Organic and conventional herbs quality reflected by their antioxidant compounds concentration, Applied Sciences-Basel, 2020, 10, s. 1-10
Reviewer 3 Report
In line 110, it was reported that in both production systems, smoked paprika samples contained significantly more zeaxanthin than the rest of the experimental samples. Could you explain what would have happened, just as speculation?
Analyzing the work of Jozsef Deli, * Péter Molnár, Zoltán Matus, and Gyula Tóth (2001) . Carotenoid Composition in the Fruits of Red Paprika (Capsicumannuum var. lycopersiciforme rubrum) during Ripening; Biosynthesis of Carotenoids in Red Paprika, Could the selfs explain in some way what would have happened, with the components after smoking? I include two works:
10.2174/1385272023373608
10.3390/antiox8100469
these expressions: change to:
In our experiment -> The experimental trials….was….
We noticed -à was noticed
We also observed t- > Was observed,
As it does not consider that the article should be exposed as a report in the first person, although it is allowed, I believe it is better to impersonally write the article.
Author Response
Reviewer no. 3
Comment 1: “In line 110, it was reported that in both production systems, smoked paprika samples contained significantly more zeaxanthin than the rest of the experimental samples. Could you explain what would have happened, just as speculation?
Analyzing the work of Jozsef Deli, * Péter Molnár, Zoltán Matus, and Gyula Tóth (2001) . Carotenoid Composition in the Fruits of Red Paprika (Capsicumannuum var. lycopersiciforme rubrum) during Ripening; Biosynthesis of Carotenoids in Red Paprika, Could the selfs explain in some way what would have happened, with the components after smoking? I include two works:
10.2174/1385272023373608
10.3390/antiox8100469….”
Authors’ response: Thank you very much for the positive evaluation of our paper and valuable advice. In fact, the proposed articles are very interesting and definitely useful in our study. Of course, all proposed articles with explain what would have happened, with the components after smoking, have been included in our study and they definitely broaden it and thus the study has become more valuable.
- Deli, J., Molnár, P. Paprika carotenoids: analysis, isolation, structure elucidation. Curr. Org. Chem., 2002, 6, 1197-1219.
- Mohd Hassan, N., Yusof, N. A., Yahaya, A. F., Mohd Rozali, N. N., Othman, R. Carotenoids of capsicum fruits: Pigment profile and health-promoting functional attributes. Antioxidants, 2019, 8, 469.
Comment 2: “these expressions: change to:
In our experiment -> The experimental trials….was….
We noticed -à was noticed
We also observed t- > Was observed,
As it does not consider that the article should be exposed as a report in the first person, although it is allowed, I believe it is better to impersonally write the article.”
Authors’ response: All these expressions have been changed as suggested by the reviewer.
The English language has been checked by professional native-speakers company American Journal Experts. We enclosed the language certificate.

Round 2
Reviewer 2 Report
The work has been corrected in many parts by the authors. Unfortunately, the question of the unidentified biological source material remains. Identifying the starting material is an indispensable requirement to allow other researchers to reproduce the results. Therefore, I believe that this parameter must necessarily be required in a journal with a high impact factor. In any case, the work is not particularly innovative and interesting.
This manuscript is a resubmission of an earlier submission. The following is a list of the peer review reports and author responses from that submission.
Round 1
Reviewer 1 Report
The introduction should be improved by including more current references to those provided, as most of those used are more than 10 years old.
The presentation of the graphs and the description, analysis and discussion of the results should be improved, the graphs are crude and without any additional information that indicates their relevance, presented in the current form is only as if the author only wanted to put the graphics because they should go there.
I suggest to emphasize the importance of the results obtained, it is important to compare them with other studies but the whole discussion is based almost exclusively on this fact. Perform a more thorough analysis of the results of the study itself and of the comparison between different groups of samples.

Reviewer 2 Report
I believe that the article is very well prepared. The theme, closely related to the purpose of the journal. The methodologies within the expected for the determinations.
I just wanted to comment, that it would be interesting to make a comparison with other sources, mainly of carotenoids, because that way the reader will be able to establish a comparison of the potential of the analyzed food.
in the end the main reason for determining compounds such as carotenoids, antioxidants, etc. are related to the value in which they can contribute to the improvement of the quality of food and consequent consumer health.
For this reason, if there is space in the manuscript, to introduce a small table with important foods containing significant amounts of carotenoids and antioxidants.
This importance is related to the frequency and quantity that these foods are consumed, therefore promoting a good effect by ingesting these bioactive substances.
Reviewer 3 Report
The aim of the work was to determine carotenoids and polyphenols in different paprika samples (sweet, hot, smoked and chili) obtained by organic and conventional production.
The work is written in an unclear and absolutely not reproducible way. The authors should have reported the agronomic methods used to cultivate peppers to allow the verification of the experiments described. The work does not present elements of originality and is written confusingly.
Reviewer 4 Report
molecules-1126675-peer-review-v1
Occurrence and determination of carotenoids and polyphenols in different paprika powder from organic and conventional production
Alicja Ponder , Klaudia Kulik, Ewelina Hallmann
Currently there is an increasing demand from consumers for natural products, to know the chemical composition, crops of origin (organic, conventional) and beneficial properties for human health. In this context, the work is relevant, it makes an important contribution, and additionally it is well structured in all its parts.
Some suggestion
- Lines 194-198, page 8
The authors write” We noticed that conventional paprika powders contained significantly more total polyphenols (432.78 195 mg/100 g DW) than organic ones (264.82 mg/100 g DW). We also observed that the concentrations of polyphenols depended on the type of product. Conventional smoked paprika and chili pepper powder contained significantly the most total polyphenols than the rest of the examined paprika samples (611.17 mg/100 g DW and 798.45 mg/100 g DW).”
Please be more specific and include a brief paragraph on how you performed the calculation of total polyphenols in the experimental section (4.4. Polyphenols analysis).
- Please include a figure in results with chemical structures of the major compounds identified. Indicate for each one of them if I identify them in conventional or organic cultivation
- In section 4.1. Analytical material preparation
For the interest of future readers of the article, please at the end of the next sentence, include the country from which the samples were purchased.
Selected paprika samples were purchased from organic and conventional shops
(Warsaw, Poland ??).
After the suggested changes, the article could be considered for acceptance in the Journal.